# Monoaminergic Systems in Flight-Induced Potentiation of Phonotactic Behavior in Female Crickets *Gryllus bimaculatus*

**DOI:** 10.3390/insects15030183

**Published:** 2024-03-09

**Authors:** Maxim Mezheritskiy, Victoria Melnikova, Varvara Dyakonova, Dmitry Vorontsov

**Affiliations:** Koltzov Institute of Developmental Biology of the Russian Academy of Sciences, Moscow 119334, Russia; v.melnikova@idbras.ru (V.M.); d.vorontsov@idbras.ru (D.V.)

**Keywords:** phonotactic behavior, female crickets, serotonin, octopamine, flight, insects

## Abstract

**Simple Summary:**

Phonotaxis is the movement of an animal towards an acoustic signal. In crickets, males generate the calling signals, thereby attracting females and other males. The calling signal is the only means of distant communication in crickets. Reception of the calling signal by the female may occur during flight, and flight often precedes copulation. We had previously found that a prior flight significantly enhanced subsequent terrestrial phonotaxis in female crickets. It is clear that some neurochemical mechanisms are responsible for the enhancement of phonotaxis after a flight. The purpose of our research was to elucidate these mechanisms. The most likely targets to test were the octopaminergic and serotonergic systems. It is known that the concentration of octopamine in the cricket hemolymph increases during flight, and that serotonin is involved in the modulation of social behavior in many animals, including insects. Using pharmacological manipulations, we show that the increase in the serotonin synthesis enhances phonotaxis. The 5-HT content in thoracic ganglia was significantly higher in flyers in comparison to the control group. The role of octopamine in phonotactic behavior is more complex and controversial. We demonstrate that octopamine suppresses rather than activates the central mechanisms associated with phonotaxis. We make suggestions about why serotonin may enhance phonotaxis and examine possible reasons for the ambiguous role of the octopaminergic system in phonotactic behavior.

**Abstract:**

We have recently shown that experience of flight remarkably enhanced subsequent terrestrial phonotaxis in females in response to the male calling song. Here, we elucidated the possible roles of octopamine and serotonin in the enhancing effect of flying on phonotactic behavior. Octopamine is known to be released into the hemolymph during flight in insects; however, the octopamine receptor antagonist epinastine did not abolish the effects of flight in our study. On the contrary, the drug significantly potentiated the influence of flying on phonotactic behavior. The octopamine receptor agonist chlordimeform, at a concentration of 2 mM, which was previously found to activate aggression in crickets, dramatically reduced the phonotactic response. However, at a 10-times-lower dose, chlordimeform produced a light but significant decrease in the time that females took to reach the source of the calling song. A similar effect was produced by octopamine itself, which hardly passes the blood–brain barrier in insects. The effect of flight was completely abolished in female crickets treated with alpha-methyl tryptophan (AMTP). AMPT suppresses the synthesis of serotonin, decreasing its content in the nervous systems of insects, including crickets. An activation of the serotonin synthesis with 5-hydroxytryptophan mimicked the effect of flight by increasing the number of visits to and the time spent in the zone near the source of the calling song. The 5-HT content in the third thoracic ganglion was significantly higher in flyers compared to the control group. In contrast, no changes in the octopamine level were observed in the third thoracic ganglion, which is known to play a crucial role in decision-making involved in intraspecific interactions. Therefore, the results suggest that although octopamine is known to be released into the hemolymph during flight, it is likely to inhibit rather than activate the central mechanisms related to phonotaxis. The weak facilitating effect of a low dose of chlordimeform can be attributed to the activation of peripheral octopaminergic receptors. Our results suggest that the serotoninergic system may contribute to the facilitation of female phonotactic behavior by flying. We suggest that both flying and serotonin enhance sexual motivation in females and, by these means, impact their behavioral response to the male calling song.

## 1. Introduction

Many animals use auditory information to choose the direction of their movement. This behavior is called phonotaxis. Insects widely use it in territorial behavior, orientation, or when searching for, choosing, or attracting a sexual partner. Several species of crickets (Orthoptera: Gryllidae) have become the most common model objects with which to study the role and mechanisms of phonotaxis in intraspecific communication as well as in predator–prey interactions. Both positive (in response to conspecific sounds) and negative (in response to the ultrasound of cricket-hunting bats) kinds of phonotaxis have been described [1,2,3]. Female crickets show positive phonotaxis to the song of a conspecific male, approaching it either in flight or by using terrestrial locomotion [4]. Phonotactic response depends on the acoustic parameters (carrier frequencies, temporal pattern, and sound intensity) of the male calling song [1,5,6].

Little is known about how different behavioral states modulate phonotaxis and what neurotransmitter systems are involved in this modulation. Presumably, hunger, danger, stress, or fertility state would affect the motivation of animals to approach the sexual partner and hence may affect the positive phonotaxis in females. Recently, an unexpected and strong behavioral modulation of the phonotaxis in female crickets (*Gryllus bimaculatus*) was revealed: a short episode of evoked flight was found to dramatically enhance the phonotactic response [7,8]. In *G. bimaculatus*, the period of natural flight usually occurs within the first week after the last imaginal molt, after which some flight muscles degrade [9,10]. Older crickets preserve the ability to activate the central pattern generator (CPG) for flying in the air stream, demonstrating so-called evoked flight. It was shown that the auditory neurons of female crickets demonstrate finer tuning to species-specific signals during evoked flight compared to the conditions of terrestrial locomotion [7]. The modulation may last longer than the flight itself; we have recently shown that a short (3 min) evoked flight remarkably enhanced subsequent terrestrial phonotaxis in female crickets; more females reached the source of the calling song and spent significantly more time near it [8]. 

Besides phonotaxis, flight is known to modulate other kinds of cricket behavior, such as intraspecific aggression in males [11,12], sensitivity to air movement detected by the cercal system and subsequent escape [12], and calling and mating behavior in males [13]. Given these findings, it was natural to presume that some mechanism of neurochemical modulation was behind the behavioral effects of flight. One of the first candidates for flight-dependent neuromodulators was octopamine (OA), since, in *G. bimaculatus*, as well as in other insects, the octopaminergic system becomes activated during flight [14]. However, only some of the above effects can be explained by the OA release. In particular, chlodimeform (CDM), an octopaminergic agonist that is able to cross the blood–brain barrier in insects, reproduces the activating effect of OA on the release of suppressed aggression in male losers, having the opposite effect on the avoidance behavior [12,15]. At the same time, the inhibitory effect of OA and its agonists on positive phonotaxis was shown in females of the house cricket, *Acheta domesticus* [16,17]. 

Serotonin, or 5-hydroxytryptamine (5-HT), may be another candidate for mediating the effects of flight. As well as flight itself, 5-HT suppresses escape behavior [12,15,18] and potentiates courtship and mating behavior in male crickets [19]. Enhanced sexual motivation was suggested to explain the effects of evoked flight on phonotaxis in female crickets [8]. However, there is no evidence of whether or how the serotonergic system responds to the activation of the flight motor program. 

The aim of this study was to test the possible mediating role of the octopaminergic and serotonergic systems in the effects of flight on the positive phonotaxis in female crickets. We tested whether the pharmacologically induced suppression of one of these systems could abolish the effect of flight on phonotaxis. We investigated whether the pharmacologically induced activation of one of these systems potentiates phonotaxis and, if it does, how similar its effects are to those of evoked flight. We also briefly evaluated the content of both monoamines in the third thoracic ganglion, known to play a crucial role in decision-making during intraspecific interactions in insects [20], in female flyers and in the control crickets. 

## 2. Materials and Methods

### 2.1. Animals

Crickets (*Gryllus bimaculatus* De Geer) were raised and kept in a laboratory colony at 25–26 °C and relative humidity 65–75% with a 12:12 light–dark cycle, fed on carrots, salad, dry Gammarus, wheat bran, and water ad libitum. Adult female crickets were taken from a mixed colony and kept in a separate colony without males for 2 weeks in order to reactivate their reproductive motivation. 

At least 12 h prior to the experiments, crickets were isolated in round plastic containers (diameter 93 mm, height 100 mm, made of opaque white plastic with a transparent perforated lid) with free access to food. This period of isolation was expected to eliminate the differences between the animals related to their social experience [21]. 

### 2.2. Behavioral Test

Neuromodulation of phonotaxis is currently being pursued by several laboratories in closely related cricket species. In the traditional approach, tethered crickets walk and phonotactically steer on the trackball system [22]. We utilize a different experimental approach. Using a flat, spacious arena, we video-track a free-roaming cricket, which is also allowed to leave its home container at will. We presume that such more-natural conditions reduce the impact of experimental stress. Previously, using this behavioral setup, we demonstrated a strong modulatory effect of evoked flight on the female positive phonotaxis [8].

The arena (150 cm × 150 cm, Figure 1A) was meant to be a novel environment for a cricket raised in a laboratory colony. The floor of the arena was made of plywood with a polyurethane matte non-slip coating, as the substrate texture was shown to affect the female cricket walking response to male calling song [23]. The walls of the arena, 20 cm in height, were made of white fabric stretched between the holders at each corner. The fabric was chosen as a sound-transparent material to hide the speaker behind it, thus avoiding the unwanted visual marker and, at the same time, decreasing the echo from the sound stimulation within the arena.

The speaker (Teac TE-T15) was located behind the left wall relative to the starting position of a cricket and was powered by an integrated amplifier (Dynavox DA-30) with a high-pass filter. The speaker, which was turned on just before placing the container with a cricket into the arena, aired the calling song of a male cricket previously recorded in the colony and looped to provide a continuous stimulation program for the time of the experiment.

The amplitude of the sound inside the arena was 85 dB SPL at 20 cm from the speaker, approximately the same level as when the sound was recorded in the colony. All sound level data in this study are given in dB SPL (sound pressure level), with a reference level of 0 dB being equal to 20 µPa.

The arena was uniformly lit by four adjustable LED panels located 1.5 m above it, providing an illuminance of ca. 280 lx at the center. A TIS DMK 23GV024 video camera (The Imaging Source, Charlotte, NC, USA) was placed 1.5 m above the arena. The movements of a cricket were recorded at 25 fps using the IC Capture software, version 2.4 (The Imaging Source). The recordings were subsequently video-tracked and analyzed in EthoVision XT 13 (Noldus, Wageningen, the Netherlands). 

At the onset of each experiment, a container with a cricket was hand-carried to the experimental arena, opened, and gently placed on the side near the wall of the arena, opening towards the center of the arena, so that the cricket could freely leave. The duration of the experiment was limited to 10 min, but it stopped earlier if a cricket escaped the arena by climbing over the fabric wall. The time spent inside a container was not considered in the analysis. The target zone of the phonotaxis was virtually outlined as a 34 × 10 cm rectangle near the speaker (“speaker zone”, SZ) (Figure 1B). To assess the phonotactic behavior, we used the following parameters: (a) the number of crickets that have reached the SZ; (b) the frequency of visits to the SZ; (c) the percentage of time spent in the SZ (from the total time spent in the arena after leaving the container); (d) the time spent to reach the speaker zone; and (e) the average speed.

### 2.3. Pharmacological Treatments

We used the octopamine receptor antagonist epinastine to suppress octopaminergic signaling. To stimulate it, we applied either octopamine or the octopamine receptor agonist chlordimeform (CDM). To suppress the serotonergic system, a false serotonin precursor, alpha-methyltryptophan (AMTP), was used. True serotonin precursor 5-hydroxytryptophan (5-HTP) was applied to increase the synthesis of serotonin. All these drugs have previously been used in *G. bimaculatus* [12,15,18,19,24,25].

All substances used in this study were dissolved in an insect saline (in mmol/L: 140 NaCl, 10 KCl, 4.76 NaHCO_3_, 2 NaH_2_PO_4_·2H_2_O, 4.2 MgCl_2_, 2.7 CaCl_2_, pH 7.0). 

For epinastine (and the respective control injection), we added 5% DMSO (dimethyl sulfoxide). Crickets were injected into the abdominal cavity. Control crickets received either saline or 5% DMSO, depending on the experiment. Detailed information on dosages and procedures is presented in Figure 1C and Table 1.

### 2.4. Evoked Flight

To induce flight behavior, the crickets were attached using thermoplastic glue to a special holder at the dorsal part of the thoracic segment and suspended for three minutes. To create the effect of wind blowing during flight, an air flow from a fan was directed at the animal, as in previous studies [8,12,13]. 

### 2.5. Serotonin and Octopamine Measurement by High Performance Liquid Chromatography with Fluorescent Detection (HPLC-FLD)

The experiment contained one experimental (*n* = 6) and one control group (*n* = 6). Females crickets in the experimental group were glued to the holder for induced flight, flew for 3 min, were detached from the holder, rested for 3 min, and then were placed on ice for 5 min before dissection. Control-group crickets were subject to similar procedures, including attachment to the holder, except no flight was induced by suspending in the air current. Control and experimental animals alternated. 

The third thoracic ganglion was quickly dissected on ice and put into the ice-cold 0.1 M HClO_4_. Ganglion tissue was homogenized with an ultrasonic homogenizer (Bandelin Sonopuls, Burladingen, Germany) at 4 °C in 60 µL of 0.1 M HClO_4_ and centrifuged at 10,000× *g* for 20 min at 4 °C. The supernatant was collected and stored at −80 °C prior to measurements of the serotonin and octopamine contents. An Agilent 1260 Infinity II HPLC system (Agilent Technologies Inc., Waldbronn, Germany) equipped with fluorescence detector (FLD) was used for the monoamines analysis.

Analytes were separated using a reverse-phase InfinityLab Poroshell 120 EC-C18 100 mm × 4.6 mm column with a 2.7 µm particle size (Agilent Technologies Inc., Germany). The column was thermostated at 30 °C. The mobile phase consisted of 0.1 M citrate–phosphate buffer, 0.25 mM 1-octanesulfonic acid sodium salt, 0.1 M EDTA, and 7% acetonitrile (pH = 2.56) (all reagents purchased from Sigma-Aldrich, St. Louis, MO, USA). The mobile phase flow rate was 1 mL/min. FLD detection was carried out at the excitation wavelength 285 nm. The emission wavelength was set at 310 nm. Peaks of monoamines were identified by the retention time relative to the standard solutions. The content of monoamines was calculated using the ratio of peak areas in the sample to the standards.

The tissue levels of monoamines were expressed as pmol per ganglion.

### 2.6. Statistical Analysis

Data analysis was performed using the PAST software (PAST: paleontological statistics software package for education and data analysis version 2.09., 2001, University of Oslo, Oslo, Norway) [26]. Statistical significance was tested using the Mann–Whitney U test, except for the number of animals that reached the speaker zone, where the Χ^2^ test was used.

## 3. Results

### 3.1. Octopaminergic Ligands and Phonotactic Behavior in Female Crickets

#### 3.1.1. Octopamine Receptor Antagonist Epinastine Potentiates the Effect of Flight

To find out whether flight affects phonotaxis through the octopaminergic system, 30 min after the administration of epinastine or 5% DMSO, we subjected crickets to flight. After the flight, the crickets were placed back into individual containers for two minutes to relieve acute stress and then placed into the experimental arena. 

In the epinastine group, the time spent in the SZ (“speaker zone”, Figure 1B) increased (*n* = 19, *n* = 19; *p* = 0.03; Z = 2.04; Mann–Whitney U-test) and the speed of locomotion decreased (*p* = 0.01; Z = 2.48; Mann–Whitney U-test) (Figure 2). No differences were found in other parameters. 

#### 3.1.2. Octopamine Receptor Agonist Chlordimeform Exerts Dose-Dependent Effects on Phonotaxis

We next tested if CDM or octopamine itself would mimic the effects of flight. The effects of CDM injection differed depending on the dose (Figure 3). Females that received 0.2 mM (*n* = 17) reached the SZ faster than the control group (*n* = 16, *p* = 0.01; Z = 2.49, Mann–Whitney U-test, Figure 3C). No significant differences were found in the other parameters. At a concentration of 1 mM, CDM decreased the speed of locomotion (*n* = 20, *n* = 21; *p* = 0.04; Z = 2.80, Mann–Whitney U-test, Figure 3I), having no effect on other behavioral parameters. At 2 mM (*n* = 15, *n* = 17), CDM caused a significant suppression of phonotaxis: only 2 out of 15 crickets in the CDM group reached the SZ compared to 10 out of 17 in the control group (Figure 3G,H). The time spent in SZ was significantly reduced (*p* = 0.01; Z = 2.49 Mann–Whitney U-test), and the locomotion speed was lower in the CDM group (*p* = 0.001; Z = 2.37 Mann–Whitney U-test, Figure 3L). 

#### 3.1.3. Octopamine (40 mM) Acts Similarly to the Low Dose of CDM

We tested the possible effects of OA injected into the abdominal cavity. Lower concentrations (10 mM, 20 mM) produced no significant changes in the cricket behavior. Higher dose (40 mM, 60 µL) resulted in reaching the SZ faster (*n* = 9, *n* = 9, *p* = 0.04; Z = 1.97, Mann–Whitney U-test, Figure 4C). No significant differences were found in the other parameters. 

### 3.2. Serotonergic Ligands and Phonotactic Behavior in Female Crickets

#### 3.2.1. Serotonin Synthesis Inhibitor AMTP Abolishes the Effect of Flight on Phonotactic Behavior

To test the possible involvement of 5-HT in the behavioral effects of flight, female crickets were injected with AMTP (171 mM, 40 µL, *n* = 10, *n* = 10) 3–4 days before flight. Alpha-methyltryptophan (AMTP) significantly reduces the content of 5-HT and its catabolite 5-hydroxyindoleacetic acid (5-HIAA) in the central nervous system, which has been demonstrated in rodents and insects [15,27,28,29]. In the AMTP flyers, phonotaxis was suppressed; not a single cricket reached the SZ (Figure 5). In the control group, 5 out of 10 crickets reached the SZ (*p* = 0.01, χ^2^ test). 

#### 3.2.2. Immediate Serotonin Precursor 5-HTP Potentiates Phonotactic Behavior

To study the effect of increased levels of 5-HT on phonotaxis, crickets (*n* = 25, *n* = 24) were injected with the immediate metabolic precursor of serotonin, 5-hydroxytryptophan (5-HTP). Lower concentration (1 mM) produced no significant changes in cricket behavior. Higher dose (10 mM, 100 µL) resulted in higher number of crickets reaching the SZ (24 of 25 versus 17 of 24, *p* = 0.02, χ^2^ test, Figure 6). In addition, they repeatedly visited SZ more times (*p* = 0.03; Z = 2.10, Mann–Whitney U-test) and spent significantly more time in it (*p* = 0.008; Z = 2.10, Mann–Whitney U-test, Figure 6C). No differences were found in other behavioral parameters (Figure 6F). 

### 3.3. Impact of Flight on Serotonin and Octopamine Content Measured by High Performance Liquid Chromatography in the Thoracic Ganglia of Female Crickets

The above data suggested a possible involvement of 5-HT in the influence of flight on phonotactic behavior. To test whether flight affects the serotonergic system in female crickets, we measured the content of serotonin in the third thoracic ganglion. This structure was chosen for analysis as previously, it was shown to be the key system controlling serotonin-dependent intraspecific attraction in locust [20]. In parallel, we also addressed the possible changes in the OA content. The 5-HT content demonstrated significant increase in flyers in comparison to the control group (Mann–Whitney U-test = 2; *p*= 0.032, Figure 7, the left group of bars). In contrast, no differences were observed in the content of OA between the control and experimental group (Mann–Whitney U-test = 15; *p*= 0.699, Figure 7, the right group of bars).

## 4. Discussion

The aim of this study was to test the possible involvement of monoaminergic signaling in the neuromodulation of phonotaxis in female crickets. Specifically, we elucidated the possible roles of OA and 5-HT in the enhancement of phonotactic behavior by flight, using drug application into the hemolymph. Some of the applied drugs, namely, epinastine, CDM, and AMTP, are known to be able to cross the blood–brain barrier and to affect not only peripheral but also central neuronal targets [12,30]. In contrast, OA hardly passes the brain–blood barrier, which functions to separate the possible peripheral and central effects of this multifunctional neurohormone [31]. Thus, an increase in the hemolymph concentration of monoamines does not necessarily reflect a similar increase in the ventral nerve cord or in the brain. 

There are numerous neurohemal sites for the OA release in insects, and the role of OA in the adjustment of the muscle metabolism for the energy-consuming flight activity is well elucidated [32,33,34]. Indeed, in *G. bimaculatus,* the level of OA in the hemolymph is known to increase tenfold during flight [14]. Pharmacological activation of the central octopamine receptors also mimicked some effects of flight on male aggressiveness in this species [12]. Although we observe some facilitatory effects of OA or its agonist, CDM, in certain doses, we rule out the possibility that OA similarly plays a major role in mediating the effect of flight on phonotaxis. The reasons for this conclusion are discussed below.

Pharmacological activation of the serotonergic system, in turn, potentiates mating behavior [19,35] and decreases anxiety [12,18]. These changes could be relevant to the enhancement of phonotactic behavior as well. Our results suggest that the serotonergic system may contribute to the influence of flying on the phonotactic behavior of female crickets. We detected a significant increase in the 5-HT content in the thoracic ganglion after the flight. Activation of the 5-HT synthesis mimics the effect of flight by increasing the time spent near the speaker producing the male calling song, while the depletion of 5-HT abolishes the effect of flight on the phonotaxis. 

### 4.1. The Dual Role of OA in the Modulation of Phonotaxis

To test the possible involvement of OA in the mechanisms of flight-induced enhancement of phonotaxis, we first tested whether epinastine abolishes the effect of flight on phonotactic behavior. Epinastine is a blocker of the histamine H1 receptors in vertebrates [36,37]. At the same time, it is a highly specific antagonist to the OA receptors in insects [30,38]. In male crickets, epinastine abolishes the effect of flight on aggressive behavior [12]. 

Surprisingly, epinastine significantly enhances the phonotactic response in flyers, which is evident in the increased time that female crickets spend near the source of the calling song (speaker zone, SZ) (Figure 2). Evoked flight itself, compared to control, produces similar effects [8]. It is noteworthy that epinastine increases phonotaxis even in flyers as in these experiments, both groups of crickets, injected with either epinastine or saline, flew prior to the behavioral test. Therefore, it is likely that epinastine inhibits the neuronal networks that suppresses the phonotactic behavior, causing its release from suppression. This suggestion is also supported by our finding of the suppressive effect of CDM on phonotaxis: the number of crickets that reached SZ and the time they spent in SZ are significantly lower after the treatment with 2 mM CDM (Figure 3G,H). 

The speed of locomotion decreases after both the CDM and epinastine injections (Figure 2F and Figure 3I,L). Thus, the change in the time spent in the SZ cannot be explained by the general change in the movement speed. It should be noted that the speed of locomotion was not affected by flight in our previous study [8].

The idea that OA may inhibit central networks involved in phonotactic behavior agrees with the recently reported OA-induced suppression of phonotaxis observed in a different species of cricket, *Acheta domesticus* [17]. In that study, OA was injected directly into the prothoracic ganglion, resulting in a reduced phonotactic response. 

At the same time, we observed a weak but statistically significant effect of a low dose (0.2 mM) of CDM on the time that crickets spent to reach the SZ. Crickets treated with a low dose of CDM accomplished it faster (Figure 3C), with no significant changes in other measured behavioral parameters. However, it is difficult to compare this effect with the influence of flight, as the time spent reaching the SZ was not evaluated in our previous study [8]. In that study, we found a significant difference in the number of flyers and non-flyers that reached the SZ, and because of this difference, the time spent reaching the SZ could not be properly measured across the two groups. 

The dose-dependent effects of CDM may indicate that there are both inhibitory and excitatory OA-dependent pathways that affect phonotactic behavior. Inhibitory ones seem to have a greater weight and are likely to rely on central nervous mechanisms. At higher concentrations, CDM successfully passes through the blood–brain barrier [12,39,40,41,42] and apparently acts on a wider range of neural circuits, which seem to be more associated with the suppression of phonotaxis. Lower doses of CDM may affect only the peripheral neurons, e.g., the auditory sensory neurons, having no effect on the central inhibitory pathways. This idea agrees with the effect of OA itself, which hardly passes the blood–brain barrier in insects [12,31]. In our study, its action is similar to the low dose of CDM (Figure 3A–C and Figure 4).

Another piece of evidence in favor of the peripheral action of low doses of CDM or OA on phonotaxis is the fact that in males of the same species, low dosages of CDM had no influence on aggression [12], which is controlled by central mechanisms. A higher dosage of CDM, 1–2 mM (the same that suppressed phonotaxis in our study), has been found to increase aggressiveness [12], suggesting the involvement of central octopaminergic mechanisms. The intermediate dose of CDM had no influence on phonotactic behavior at all, which may suggest that peripheral and central mechanisms were balanced. To conclude, we suggest that activation of the peripheral OA receptors slightly enhances the phonotaxis, while the central action of OA suppresses it. 

#### 4.1.1. Behaviors Activated by Octopamine Can Interfere with Positive Phonotaxis

Several behaviors, which are known to be activated by OA in insects, may compete with sexual motivation and thus reduce phonotaxis. In *G. bimaculatus*, octopamine potentiates anxiety, escape responses, and aggression [12]. In *Drosophila melanogaster*, octopaminergic neurons activate post-mating behavior and decrease the receptivity of females to male courtship attempts [43,44]. Post-mating behavior is also characterized by increased intraspecific aggression in female *Drosophila* [45,46,47]. Intraspecific aggression was reported to decrease the courtship behavior [35,48] and mating success in crickets [49]. Therefore, the spectrum of behaviors activated by OA allows us to presume at least an indirect negative effect of OA on the female phonotaxis, which agrees with our hypothesis that central octopaminergic receptors are involved in the suppressive control of phonotactic behavior. 

#### 4.1.2. Is There Evidence for a Facilitating Effect of Octopamine on Insect Hearing?

In our experiments, the decrease in time to reach the SZ could be due to the activation of octopaminergic receptors in the auditory system. In crickets, the first-order interneuron, called AN1, is known to be associated with positive phonotaxis to intraspecific acoustic cues [50]. Both primary receptor neurons and the AN1 interneuron respond to auditory stimuli in a fairly wide range of stimulus characteristics. Nevertheless, the signs of selectivity for species-specific calls were revealed even at these early stages of signal processing. In the brain, a more pronounced selectivity of species-specific acoustic parameters appears [50]. Whether and how these neurons are modulated by OA or 5-HT remains to be studied.

In the cricket *G. bimaculatus*, finer tuning of auditory neurons to species-specific signals was revealed during flight compared to terrestrial locomotion [7]. The in-flight modulation of the auditory system is consistent with the results obtained in another insect, the blowfly, which demonstrated a change in the tuning of motion-sensitive visual neurons during flight [51]. This effect of flight was reproduced by the application of CDM, suggesting the role of the octopaminergic system in the in-flight modulation of sensory systems. OA indeed exerts neuromodulatory effects on different insect sensory systems [52]. 

Recent studies in mosquitoes demonstrated that OA modulated both the mechanical and neuronal frequency tuning of the auditory system [53,54]. The activity of the octopaminergic system was found to be correlated with the swarming hours, when a male mosquito relies on its audition to detect and pursue a female [53], while the neuronal effects of OA and CDM were different in male and female mosquitoes [54]. 

The activity of the most-studied octopaminergic neurons in the insect nervous system, dorsal medial unpaired (DUM) neurons, is, in turn, affected by various types of sensory cues, including auditory ones [55]. In crickets, DUM neurons are suggested to participate in the modulation of sensorimotor integration [56].

To conclude, insect sensory systems are likely to be constantly modulated by OA, especially during flight. We cannot exclude that such modulation lasts for some time after the flight is over, which could be the case in our experiments.

### 4.2. Serotonin Potentiates Phonotaxis

Octopamine and serotonin often have opposite effects on insect behavior [57]. Therefore, after the octopaminergic system, the serotonergic system was the next candidate for testing the possible impact on phonotaxis. For the first time, to our knowledge, we found evidence for the involvement of serotonergic mechanisms in the behavioral regulation of phonotaxis in crickets. This finding agrees with recent observations in mosquitoes, which demonstrated a clear dependence of phonotactic response upon the 5-HT level and the state of the 5-HT receptors [58]. In our study, injection of the immediate serotonin precursor 5-hydroxytryptophan (5-HTP) significantly increased the number of visits to and the time that crickets spent in SZ. This effect is remarkably similar to the action of evoked flight on the phonotactic response [8]. Moreover, the effect of evoked flight on phonotaxis was abolished in female crickets pretreated with the 5-HT-depleting drug, AMPT (Figure 5). AMPT completely prevented the phonotactic response, which was clearly below the control level observed in non-flyers. Mosquitoes *Aedes aegypti* treated with AMTP similarly demonstrated a significant reduction of phonotactic response [58]. The combined evidence suggests that 5-HT positively regulates phonotactic behavior and may be involved in the mechanisms of flight-induced potentiation of phonotactic behavior in crickets. We suggest that flight potentiates phonotaxis by enhancing the serotonin release in the CNS rather than by recruiting other serotonergic mechanisms like de novo receptor expression, etc.

#### 4.2.1. Does Flight Change the State of the Serotonergic System?

We provide, to our knowledge, the first evidence that the content of serotonin in the nervous system is sensitive to flight in *G. bimaculatus*. Namely, it was significantly increased in the third thoracic ganglion of female crickets that flew and then rested for 3 min. This protocol repeats the one used for behavioral experiments on flying (here in and [8]). Interestingly, an increase in the serotonin content in the thoracic ganglion was observed in the locust *Schistocerca gregaria* during swarming [20,59].

In a different insect, the large carpenter bee *Xylocopa appendiculata*, the levels of 5-HT and 5-HTP were significantly higher in the brains of males collected at the time related to their territorial flights [60]. It is not clear, however, whether this difference can be attributed to the effect of flight itself. The vice versa relationship, namely, the influence of the 5-HT level on the locomotor behaviors in insects, is better elucidated.

Normal functioning of the serotoninergic system is required for insect movement [61,62]. Severe perturbations in the 5-HT level result in an inability to fly [61] and decreased locomotor activity in various insect species [15,62,63], including *G. bimaculatus* [15]. At the same time, acute activation of serotoninergic interneurons and the 5-HT release seems to elicit the termination of flying. This early hypothesis, proposed in the studies of the moth, *Manduca sexta* [64], was supported by the results of recent detailed investigations of the 5-HT neuromodulation in *D. melanogaster* [62,65]. Several reports show inhibitory effects of 5-HT-reuptake inhibitors, 5-HT itself, or methiothepin, a non-selective antagonist of the 5-HT receptors [66,67], on flight performance in mosquitoes [68,69]. It seems likely that the serotonergic system may be responsible for the termination of flight as a negative feedback loop, which provides a switch to slower types of movement. This hypothesis needs experimental verification by direct measurements of the 5-HT levels in the nervous system during flight and after its termination. 

#### 4.2.2. Serotonin May Affect Phonotaxis via Changes in the Motivational State

The influence of 5-HTP on the attractiveness of the conspecific calls may be explained by the change in the motivational state of crickets. Serotonin is known to impact motivations in different phyla, from protostomes to humans [70,71,72,73]. It has been suggested that an increase in the 5-HT content can cause a switch in behavioral response to various external signals from “no” to “yes”, i.e., from ignorance and avoidance to approaching and investigation [24]. In the locust *Schistocerca gregaria*, 5-HTP caused a general change of motivational state towards conspecifics from avoidance to approach, leading to gregarization and swarming [20]. Later, however, the opposite results were obtained from different locust species (*L. migratoria*), which allowed the authors to conclude that serotonin does control approach and withdrawal responses, but in a species-specific manner [74].

In *G. bimaculatus*, 5-HT is involved in the negative control of aggression [25] and anxiety [12,18]. 5-HTP was found to affect male sexual behavior by decreasing the refractory period after copulation [19] and to activate calling song production and courtship behavior [13,35]. 

In our experiments, changes in the female phonotaxis induced by the flight or the 5-HTP injection (Figure 6) can also be explained by the increased sexual motivation. The lower number of crickets that escaped the experimental arena (thus showing no interest in the calling song) after the evoked flight or 5-HTP injection agrees with this hypothesis. 

#### 4.2.3. Serotonin and Hearing in Insects

The effects of 5-HT on phonotaxis can also be explained by the activation of the 5-HT receptors in the auditory system. To our knowledge, there are no studies on the role of serotonergic regulation in cricket hearing. However, mosquitoes possess serotonergic innervation of the primary auditory neurons of the Johnston organ (JO). This innervation is believed to be involved in the efferent control of hearing [75,76], and several types of 5-HT receptors (5-HT_1_, 5-HT_2_, and 5-HT_7_) are expressed there [58]. Thoracically injected 5-HT caused an increase in the mechanical tuning frequency of the JO, reversed by the injection of methiothepin [58]. The opposite effect, a clear decrease in the mechanical tuning frequency of a male mosquito JO, was observed after depleting the 5-HT content via AMTP. The authors of [58] concluded that changes in the 5-HT level alter the phonotactic profiles of male mosquitoes via modulation of their hearing function. 

Given these findings, the direct effects of 5-HT on the auditory system cannot be excluded in crickets, which may act cooperatively with the central serotonergic mechanisms.

### 4.3. Conclusions. Possible Roles of 5-HT and OA in the Flight-Dependent Modulation of Female Phonotactic Behavior

Our results suggest that although the hemolymph levels of OA are known to increase in crickets during flight, OA is likely to inhibit rather than activate the central mechanisms related to phonotaxis. The small positive effect of a low dose of CDM may be attributed to its action on the OA receptors in the peripheral auditory system. In contrast, we observed a significant increase in the content of 5-HT in the third thoracic ganglion after the flight; pharmacological activation of the serotonergic system enhanced phonotaxis, while 5-HT depletion prevented the flight-induced phonotaxis. Thus, we suggest that central 5-HT may contribute to the positive effect of flight on female phonotactic behavior. 

## Figures and Tables

**Figure 1 insects-15-00183-f001:**
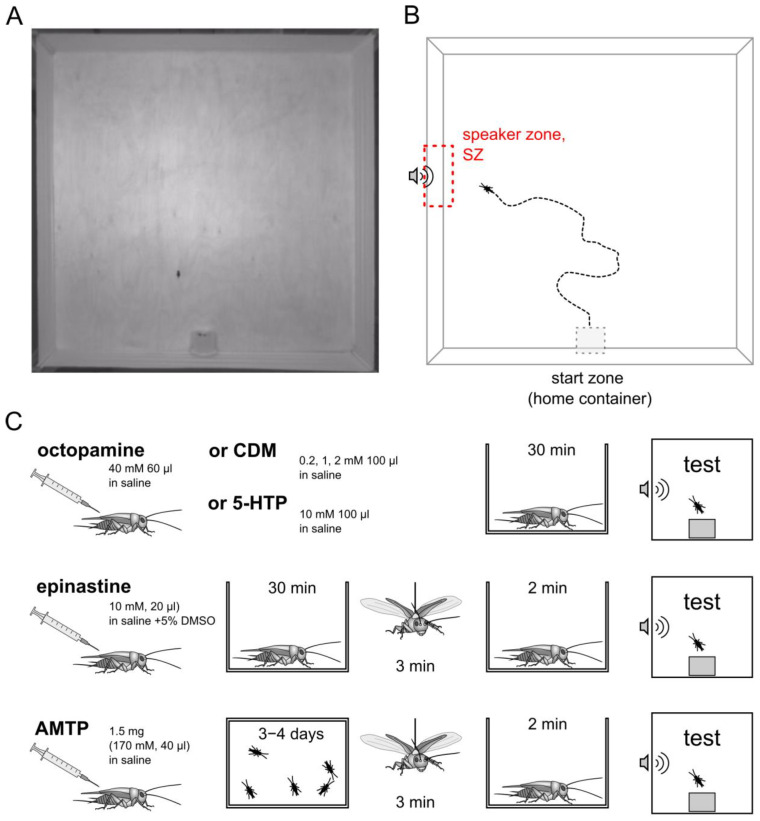
Experimental protocol. (**A**) Snapshot from a recorded video showing the arena with a plastic container put in it and a cricket that emerged from the container. (**B**) Schematic diagram of the arena, showing a typical track of phonotactic behavior towards the source of the calling song. (**C**) Schematic representation of the protocol of pharmacological experiments showing the sequence of procedures from injection to test, left to right.

**Figure 2 insects-15-00183-f002:**
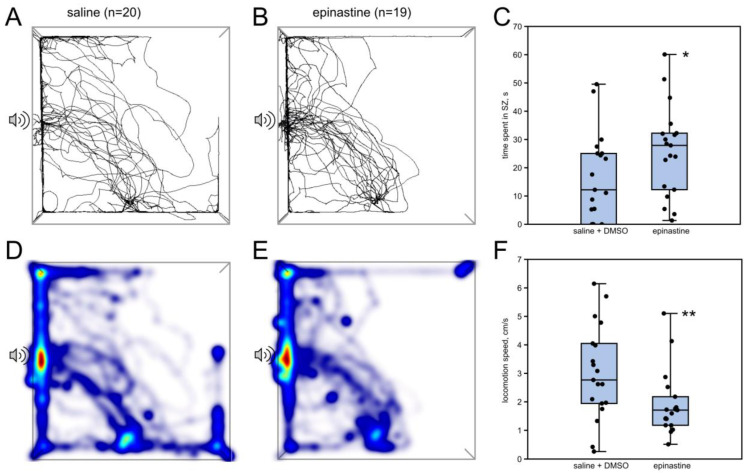
Effects of epinastine on the phonotaxis in female crickets that were previously subjected to induced flight. (**A**,**B**) Superimposed individual tracks of crickets in the control and epinastine groups, respectively. (**D**,**E**) The same, shown as heatmaps, which better represent the integral time spent in different parts of the arena, color indicates relative integral time, from blue (minimum) to red (maximum). (**C**) Epinastine increased the time spent in SZ. (**F**) Epinastine decreased the locomotion speed. Asterisks indicate the level of statistical significance according to the Mann–Whitney U test: * *p* ≤ 0.05, ** *p* ≤ 0.01.

**Figure 3 insects-15-00183-f003:**
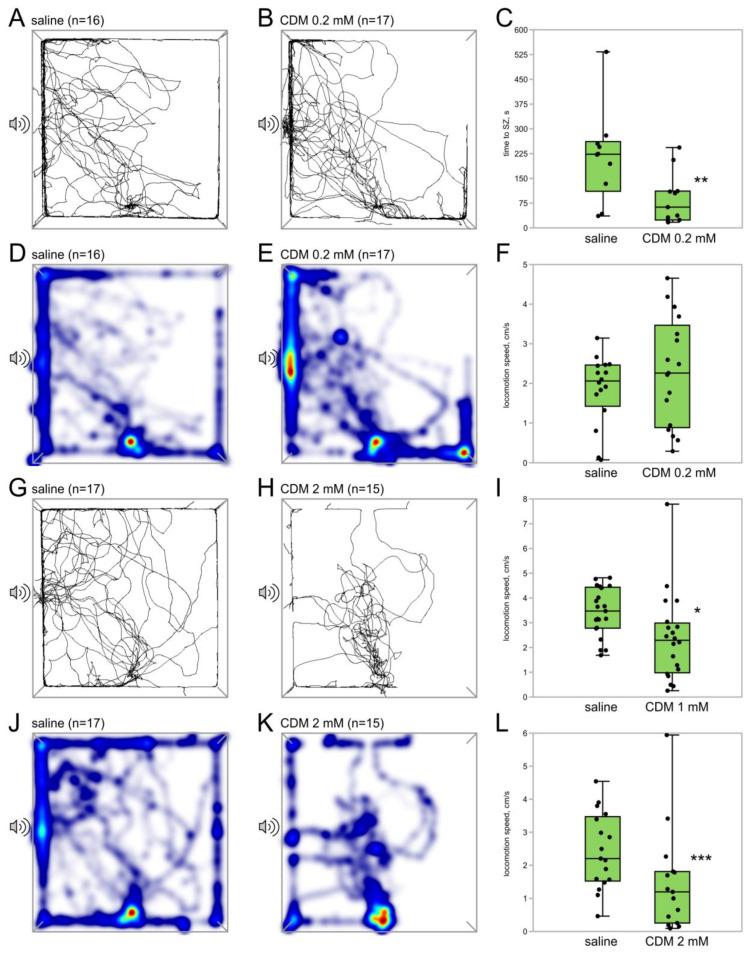
Effects of different doses of chlordimeform on the phonotaxis in female crickets. (**A**,**B**,**G**,**H**) Superimposed individual tracks of crickets in the control (**A**,**G**) and CDM groups with different concentrations of CDM injected: 0.2 mM (**B**) and 2 mM (**H**). Note that significantly less tracks enter the area near the speaker in the 2 mM CDM group (**H**). (**D**,**E**,**J**,**K**) The same, shown as heatmaps, which better represent the integral time spent in different parts of the arena, color indicates relative integral time, from blue (minimum) to red (maximum). CDM 0.2 mM decreased the time to reach the SZ (**C**) and did not affect the speed of locomotion (**F**). CDM 1 mM and 2 mM decreased the locomotion speed ((**I** and **L**), respectively). Asterisks indicate the level of statistical significance according to the Mann–Whitney U test: * *p* ≤ 0.05, ** *p* ≤ 0.01, *** *p* ≤ 0.001.

**Figure 4 insects-15-00183-f004:**
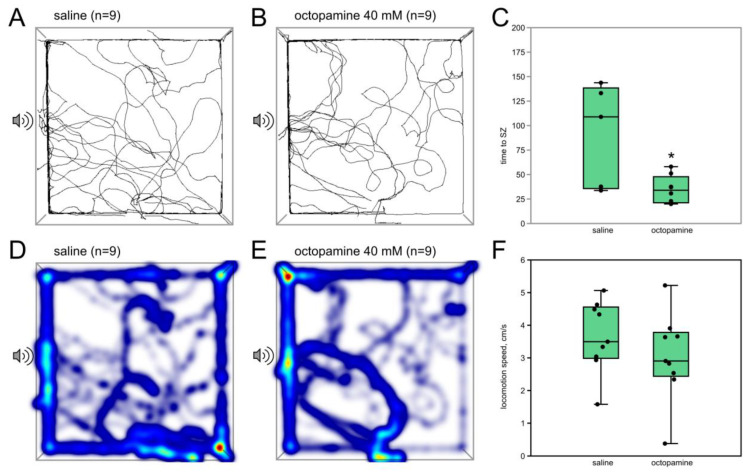
Effects of octopamine on the phonotaxis in female crickets. (**A**,**B**) Superimposed individual tracks of crickets in the control and octopamine groups, respectively. (**D**,**E**) The same, shown as heatmaps, which better represent the integral time spent in different parts of the arena, color indicates relative integral time, from blue (minimum) to red (maximum). Octopamine decreased the time to reach SZ (**C**) and did not affect the locomotion speed (**F**). An asterisk indicates the level of statistical significance according to the Mann-Whitney U test: * *p* ≤ 0.05.

**Figure 5 insects-15-00183-f005:**
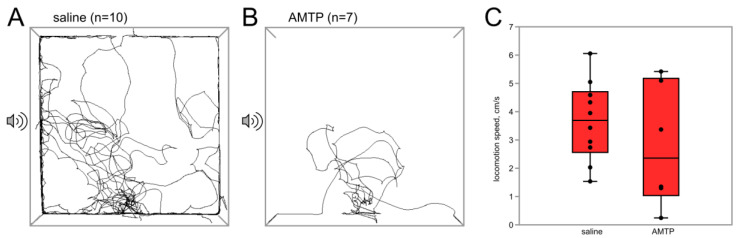
Effects of AMTP on the phonotaxis in female crickets that were previously subjected to induced flight. (**A**,**B**) Superimposed individual tracks of crickets in the control and AMTP groups, respectively. Note that not a single cricket entered the area near the speaker in the AMTP group. (**C**) The locomotion speed was not significantly affected by AMTP.

**Figure 6 insects-15-00183-f006:**
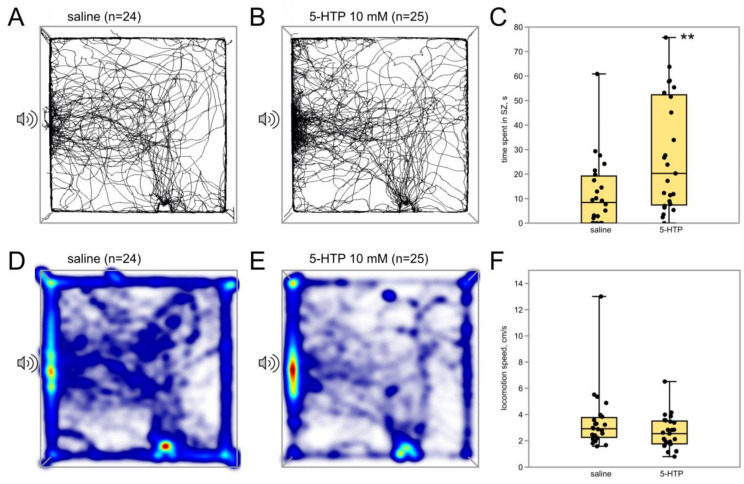
Effects of 5-HTP on the phonotaxis in female crickets. (**A**,**B**) Superimposed individual tracks of crickets in the control and 5-HTP groups, respectively. (**D**,**E**) The same, shown as heatmaps, which better represent the integral time spent in different parts of the arena, color indicates relative integral time, from blue (minimum) to red (maximum). Note that crickets tend to spend more time near the speaker in the 5-HTP group (**E**) compared to control, which is also shown in (**C**,**F**). The locomotion speed was not affected by 5-HTP. Asterisks indicate the level of statistical significance according to the Mann–Whitney U test: ** *p* ≤ 0.01.

**Figure 7 insects-15-00183-f007:**
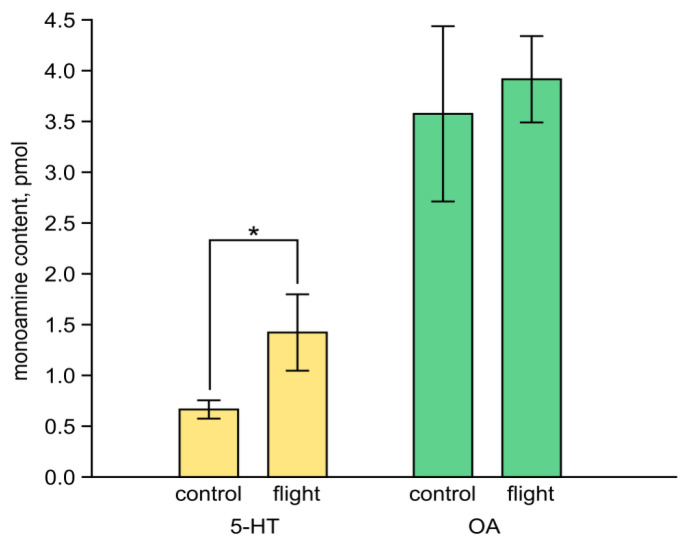
Serotonin and octopamine content in the third thoracic ganglion in flyers and non-flyers. Data are expressed in pmol per ganglion and represent the Means + SEM, *n* = 6. An asterisk indicates the level of statistical significance according to the Mann-Whitney U test: * *p* ≤ 0.05.

**Table 1 insects-15-00183-t001:** Pharmacological treatments of crickets prior to behavioral tests.

Agent	Concentration	Volume of Injection	Vehicle	Remarks
Octopamine hydrochloride (Sigma)	40 mM	60 µL	saline	Crickets were injected 30 min prior to testing. Octopamine acted for at least 1.5–2 h after injection.
Chlordimeform (CDM, the octopamine receptors agonist) (Sigma)	0.2 mM, 1 mM 2 mM	100 μL	saline	Crickets were injected 30 min prior to testing. Chlordimeform acted for at least 1.5–2 h after injection.
Epinastine, a specific antagonist of insect octopamine receptors (Sigma)	10 mM	20 μL	5% DMSO	Crickets were injected 30 min prior to the flight procedure. After the flight and before being placed in the experimental arena, the crickets were placed in individual containers for two minutes to rest. After a short rest, the crickets were placed in the experimental arena. Epinastine acted for at least 1.5–2 h after injection.
5-hydroxytryptophan (5-HTP), immediate metabolic precursor of serotonin	10 mM	100 μL	saline	Crickets were injected 30 min prior to testing. 5-HTP acted for at least 1.5–2 h after injection.
Alpha-methyltryptophan (AMTP), serotonin synthesis inhibitor, the «false serotonin precursor»	171 mM	40 µL *	saline	Crickets were injected 3–4 days prior to the flight procedure. After the flight and before being placed in the experimental arena, the crickets were placed in individual containers for two minutes to rest. After a short rest, the crickets were placed in the experimental arena.

* One dose contained 1.5 mg of AMTP.

## Data Availability

Data are contained within the article.

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
