# Peer review of "Monoaminergic Systems in Flight-Induced Potentiation of Phonotactic Behavior in Female Crickets Gryllus bimaculatus"

_insects, 2024, doi:10.3390/insects15030183_

Round 1
Reviewer 1 Report
Comments and Suggestions for Authors
I read the paper with great interest from the point of view of someone in the field of auditory physiology. I found the paper easy to read and logical. In additionto the new findings and their interpretation, I also appreciated the shortcomings and limitations of the outcomes that were identified by the authors. I spotted a few places whwre the text could be made clearer. Unfortunately i have been taken ill during the review and may have missed important points identified by other reviewers. In view of this i am happy to look at a revised manuscript to see if I support suggestions for change. In my current view, any reservations I had initially with the interpretation of the outcomes, especially concerning central versus peripheral mechanisms in the control of phonotaxis, were dealt with by the authors.

Comments on the Quality of English LanguageEnglish is excellent with few suggestions for change
Author Response
Thank you for your positive evaluation of our work and for your comments. All corrections have been made to the text of the article
Reviewer 2 Report
Comments and Suggestions for Authors

Comments on the Quality of English LanguageAuthor Response
Thank you for your comments. Please see the attachment.

Reviewer 3 Report
Comments and Suggestions for Authors
In this paper, M. et al. are exploring the role of bioamines in modulation of phonotaxis by recent flight. Their previous work introduced recent flight activity as a potentiator in the attraction behavior of female Gryllus bimaculatus crickets to male mating calls. Timescales involved, i.e. prolonged modulation, justify their hypothesis in testing bioamines in this paradigm. Octopamine and serotonin have been shown to mediate modulation of sensory and reward circuits, contributing to behavioral states in numerous insect studies.
The experimental design, introduced clearly in Figure 1, is straightforward in implementing established pharmacological interventions with well-known antagonists and agonists for these bioamines. The rest of the figures analyze the impact of their interventions via several behavioral parameters, such as time spent in SZ (stimulus zone, proximity to the stimulus), latency to SZ, locomotion and include heatmaps and trajectories. While a more dynamic trajectory analysis could have yielded new insights in the paper, these parameters are certainly sufficient to describe phonotaxis behavior.
Figures 2 – 4 explore the contribution of octopamine in phonotaxis. Octopamine antagonist epinastine injected crickets were slower than controls, but albeit spent longer duration in SZ. (While it is not trivial to disentangle reduced locomotion and time spent in active zone, reduction is speed does not seem to be drastic enough to explain longer times spent in SZ.) In contrast, gain-of-function analysis showed dose dependency. Low-concentration agonist CDM and octopamine applications led to higher attraction to SZ (shorter latency). Higher CDM concentration, however, reduced lower speed and phonotaxis.
Figures 5-6 explore the contribution of serotonin in phonotaxis. Antagonist AMPT strongly reduced the cricket exploratory behavior. Since locomotion is not significantly reduced compared to the controls, authors suggest attraction-to-male calls are reduced in the absence of serotonergic activity. Unfortunately, only seven repetitions are reported here, markedly less than the rest of the paper. More repetitions are needed to show that AMPT does not influence locomotion and therefore I would strong suggest to perform them. Serotonin alone, in return, created the strongest phenotype in the paper, and phonotaxis is strongly enhanced.
Taken together, these figures tell a story where octopamine has an ambiguous role, and serotonin is a strong candidate as the chemical for flight-induced potentiation of phonotaxis.
The most significant caveat in the paper lies with establishing the link between the candidate neuromodulators and flight-induced potentiation. In gain-of-function experiments, agonists were applied without evoked flight, thus effectively acting as a substitute. In loss-of-function experiments, systemic injection of antagonists would lead to preventing the expression of evoked flight’s impact on phonotaxis. Establishing sufficiency and necessity in this way is a common method in behavioral pharmacology. However, a significant set of experimental controls are missing. (Octopamine and) Serotonin can impact via phonotaxis independently of evoked flight. To strengthen the proposed link between these bioamines and previous behavioral experiments, antagonists should be applied in the absence of evoked flight. If same results were to be acquired in such absence, the major claims in the paper should be in doubt.
Additional notes:
- - Between figures, behavioral readouts (time spent, latency, locomotion) are not consistently used, which hinders the manuscript's overall readability. Additionally, in the text, entry times and number of crickets entered SZ per condition were also reported, however, they were not depicted in the figures.
- - Octopamine is involved in several crucial insect behaviors, including controlling locomotion levels and exploration-exploitation switch. Interfering with these behaviors under different experimental designs could lead to such ambiguous results as observed by the authors. Since octopamine is broadly expressed, one way to reduce such ambiguity would be using RNA interference to target specific octopamine receptors instead of pharmacology. (Due to availability of Gryllus bimaculatus genome, designing efficient RNAis should be straightforward). Future use of RNAi against serotonergic receptors will be also a strong supporting evidence and reveal more in investigating its involvement in phonotaxis modulation.
- - Authors use only one stimulus condition. A psychometric function could be more telling by charting female crickets’ phonotaxis to various stimulus conditions (call amplitude/frequency). Depending on how such a curve changes with pharmacological intervention, for example, whether serotonin impinges on sensory or central circuits can be deduced.
Comments on the Quality of English LanguageApart from few typos (including one in the abstract), the text is clear.
Round 2
Reviewer 2 Report
Comments and Suggestions for Authors
The authors have satisfactorily taken all comments into account. I have no further objections.
Reviewer 3 Report
Comments and Suggestions for Authors
Dear Authors,
Thank you for providing the HPLC data which strongly strengthens your hypothesis on the possible link between flight and phonotaxis.
I appreciate also your wording in reporting these results, as the increase in serotonin at third thoracic ganglion is a correlation. However, I will concede that identifying and mapping these serotonergic circuits on flight potentiation is a considerable effort and best left to future studies.
Kind regards,